# Search for the Theological Grounds to Develop Inclusive Islamic Interpretations: Some Insights from Rationalistic Islamic Maturidite Theology

**Galym Zhussipbek [1],\* and Bakhytzhan Satershinov [2]**

[1] Department of Social Sciences, Suleyman Demirel Atindagy Universitet, Kaskelen,
Almaty 040900, Kazakhstan
[2] Department of Religious Studies, Institute of Philosophy, Political Science and Religious Studies,
Almaty 050010, Kazakhstan; bakhyt-zhan@mail.ru
\* Correspondence: galym.zhussipbek@gmail.com

**Abstract:** Inclusive religious interpretations accept that a salvation beyond their teachings can be found. Whether Islam accepts inclusive religious interpretations or not, constitutes one of the most debated issues related to Islam in our days. In this paper it is argued that, although al-Maturidi's views can hardly be described as inclusive, the dynamic rationalistic Maturidite theology (so-called Maturidite "software") may help produce inclusive Islamic interpretations. In addition to the key two principles of rationalistic Maturidite theology, especially the Maturidite understanding of the fate of people not exposed to divine mission, may be understood as accepting that the people may reach faith which is similar to valid through their reason, and they may be saved hereafter, although they do not believe in Islamic teachings in a strict sense. This Maturidite position can also be used to justify the inclusive understanding of Quranic verses. By and large, Maturidite theological views analyzed in this article can be seen as factors laying the grounds to develop inclusive Islamic interpretations.

**Keywords:** Islam; rationalistic Islam; Maturidite theology; inclusivity; exclusivity; inclusive religious interpretations; Bigiev

---

## 1. Introduction: Exclusivity–Inclusivity Debate and Islam

Although exclusivity and inclusivity in religious traditions have been interpreted in different ways (Bakar 2009, p. 6), the most popular interpretation of exclusivity and inclusivity pertains to the issue of the possibility of post-humus salvation outside one's own belief and faith system. While an inclusive religious doctrine accepts a possibility to enter Heaven for non-believers in its teachings, an exclusive religious belief denies such a possibility (Bakar 2009, p. 6). In other words, inclusivism accepts that there are more than one way to salvation; therefore, inclusive religious doctrine accepts the opportunity of salvation beyond its teachings. In contrary, exclusive religious doctrine denies the opportunity of such salvation (Kamali 2011, p. 715).

Exclusivity or inclusivity are not specific to any particular religion. Both the adherents of exclusive and inclusive interpretations can be found among the followers of any religion. As such, any religion claims that only its followers can be saved, and it is natural that in any religion exclusivists tend to outnumber believers in inclusivity. By and large, religious inclusivism is not welcomed by many believers, since they tend to believe that they would be in danger of losing their own faith, if they were to admit the opportunity of salvation beyond their own religious traditions (Bakar 2009, p. 6). That being said, religious inclusivism is one of the most controversial, even taboo, issues in many religions.

Mohammad H. Khalil (2012), in his comprehensive analysis of the views of different Muslim scholars, challenges the dominant perception about Islam's attitude to exclusivity–inclusivity debate,

which assumes that there is a particular, tended to be exclusivist, Islamic orthodoxy in the issue of the soteriological fate of others. Khalil explores the variety of inclusive and exclusive perspectives developed throughout Islamic history by different scholars, such as Al-Ghazali, Ibn Taymiyya, Al-Jawziyya, Ibn Arabi, and Rashid Rida.

Nonetheless, in our days, Islam is generally accepted as an extremely exclusivist, moreover intolerant, religion. Islam's intolerance of others is explained as stemming from its exclusivist interpretations (Bakar 2009, p. 6). Exclusivist understandings of the primary Islamic source, Quranic verses, became dominant in traditionalist Muslim scholarship and society. For example, the Muslims tend to narrowly interpret and literally understand the following Quranic verses, where the message of universality and inclusivity can also be inferred, "True religion before Allah is I/islam (submission to His will)" (3:19); "If anyone seeks a religion other than (I/islam) complete devotion to God, it will not be accepted from him . . . " (3:85); and "this day I perfected your religion for you, completed My favour to you, and chose I/islam to be your faith" (5:3).

We think that underdevelopment of Islamic discourses on religious inclusivism is not a problem of Islam as such, basically, it is improper to view Islam and any other religion as monolithic and static. In essence, any religion is a social phenomenon, theology is a human construct, and religious views of the believers are formed and transformed by their environments, education, and the information they are exposed to.

We argue that rationalistic Islamic theology, by which in this article we denote Maturidism, may have a potential to produce inclusive Islamic interpretations. One of the paradoxes of Islamic theology and history is that, despite Maturidism being nominally accepted as one of the two Sunni-Islamic schools in creed and theology, in fact, it has been overshadowed for many centuries by Asharism (Abdallah 1974, p. 7) and dynamic rationalistic Maturidite epistemology was largely lost. However, it is a matter of fact that al-Maturidi's views per se about other religions can hardly be categorized as inclusive.

In this article we argue that it is needed to move from exclusivism-inclined al-Maturidi views to inclusivism-inclined Maturidite rationalistic principles. In other words, while the limits of al-Maturidi's inclusiveness need to be acknowledged, the rationalistic Maturidite theology, specifically its two principles, which are briefly touched in this article, can be used to produce inclusive Islamic understandings. Additionally, the other Maturidite theological views discussed in this paper, such as the Maturidite understanding of the fate of people who did not hear of Prophetic mission, Maturidite interpretations of the foundation of faith and the idea of subjectivity of faith, Maturidite position on predestination, and, finally, the acceptance of the notion of irja as a rationalistic method of understanding religion, can be seen as factors laying the grounds to develop inclusive Islamic interpretations.

## 2. From Exclusivism-Inclined Al-Maturidi Views (Maturidite "Hardware") to Inclusivism-Inclined Maturidite Rationalistic Principles (Maturidite "Software") to Produce Inclusive Islamic Interpretations

In general, the concepts of rationalistic and rationalism in Islam are mostly associated with the Mutazilate school (Zhussipbek and Nagayeva 2019, p. 348), which, in fact, in some key theological issues, accepted irrational, even extremist views. For example, similar to some radical teachings, the Mutazilate school assumes that actions are acts of faith (Haj 2009, p. 222). Though Maturidism has been conventionally acknowledged along with Asharism as one of the two Sunni-Islamic schools in creed and "kalam" (dialectical theology), in practice its epistemology turned out to be lost for centuries. In other words, while the creed developed by al-Maturidi (the eponym of Maturidite theology) was not lost, furthermore, it became a strictly dogmatic and ossified teaching, the rationalistic Maturidite theology was put into oblivion. As such, al-Maturidi was a thorough dialectician and his main endeavor was to find out the philosophical bases for his theological interpretations (Ali 1963, pp. 245–46). Moreover, al-Maturidi can be depicted as an academic scholar trained in many

disciplines related to complex theological debates but not as a traditional religious author in narrow sense. Regretfully, most of his works were lost, only two books survived (Rudolph 2016, p. 87).

We argue that it is necessary to differentiate between theologically rationalistic and dynamic Maturidite theology, which we tentatively call "software", on the one hand, and Maturidite creed, which is accepted by mainstream-traditionalist Hanafi Muslims as ossified, even dogmatic teaching, which we tentatively depict as "hardware", on the other (Zhussipbek and Nagayeva 2019, p. 354).

Al-Maturidi's views about other religions can be placed under the category of "hardware". The analysis of al-Maturidi's "magnum opus", Kitab at-Tawhid (Book of Oneness of God) can show the exclusivity of al-Maturidi's views, for example, Rifat Atay (1999, pp. 28–32), in his comparative research, showed that the writings of al-Maturidi ought to be categorized as religiously exclusivist.

As such, the focal-point of "kalam" science (al-Maturidi was a scholar of "kalam") is the refutation of heretical Islamic sects (Haydar 2016, p. 54) and other religions deviated from monotheism. Rudolph reminds that, after analyzing al-Maturidi's "Kitab at-Tawhid", it appears that the list of his theological opponents is long, and it contains several religions beyond Islam, as well as a number of Islamic sects and thinkers (Rudolph 2015, p. 149). Al-Maturidi extensively refuted the beliefs, and defended the correct conception, of monotheism ("tawhid") against Mutazilates (al-Maturidi 1970, pp. 86–92, 215–16, 236–56), Ismailis (al-Maturidi 1970, pp. 93–100), and other heretical Muslim groups such as the Karramites, Kharijits, and the followers of Jabriya and Qadariya (al-Maturidi 1970, pp. 225–28).

Al-Maturidi attacked the views of dualists of different groups such as the Zoroastrians, Marcionites, and Manichaeans (al-Maturidi 1970, pp. 157–72). As Rudolph indicates al-Maturidi's critique of dualists was not based on a general goal, as other comparable Islamic polemics, but rather had a multi-layered structure (Rudolph 2015, p. 178). The significant presence of Manichaeans and Zoroastrians in Transoxania, where al-Maturidi lived and worked, could induce him to dwell on these religions more than others to claim supremacy for Islamic theology (Rudolph 2015, p. 178). He also refuted the Ancient Greek philosophical legacy which he conceptualized as "Dahriya", which included Aristotle, the determinists ("ashab al-tabai") (al-Maturidi 1970, pp. 113–18, 121–23).

While Al-Maturidi was willing to employ the agreement of Muslims with other monotheists, such as Jews and Christians[1], he criticized and refuted the Christian and Jewish religious views. Nonetheless, it is noteworthy that al-Maturidi placed his refutations of the Christian positions in the section of his book dealing with Prophethood, but not in the section about God (al-Maturidi 1970, pp. 210–15). Additionally, al-Maturidi's rebuttal of the Christian views in the "Kitab at-Tawhid" was not put forward in the manner of a participation in some independent, ongoing tradition of Islamic polemic against the Christians (Griffith 2011, p. 639).

As such, al-Maturidi's views per se about other religions, for example, Christianity and Judaism, can hardly be described as inclusive (nonetheless, the Turkish researcher of al-Maturidi's legacy, Hanifi Ozcan, suggested that al-Maturidi provided a good example of inclusivist religion between Abrahamic religions (Ozcan 1995; Atay 1999, p. 29)). Nonetheless, despite the exclusivism-inclined al-Maturidi views (Maturidite "hardware"), we argue that using a broader perspective, dwelling on the rationalistic Maturidite theology, specifically on its two principles, may lead the Muslims to produce inclusive Islamic understandings.

In our view the rationalistic Maturidite epistemology is premised upon two main principles. The first principle is the belief that reason can find what is good and what is bad independently from revelation (al-Maturidi 1970, pp. 9–10). As such, reason and revelation both occupy a prominent place in the theological system of al-Maturidi (Ali 1963, p. 263). Nonetheless, al-Maturidi assumed that in many cases only reason can reveal the truth (Ali 1963, p. 263). As such al-Maturidi accepted the rationality of ethical norms, although not in absolute terms (al-Maturidi 1970, pp. 9–10). Therefore, we

---

[1]　　As an anonymous reviewer reminded.

conceptualize the second principle as the belief that God does not order to do what is known by reason as bad or evil.

Maturidite theology teaches that human rational knowledge extends over various domains and, in distinction to the Ashari school, human rational knowledge encompasses ethical norms (Rudolph 2015, p. 300). Therefore, contrary to the Asharite theology, which accepts what is good and what is bad based only on God's will, Maturidites and Mutazilates accept that the basis of God's commands are objective standards (Deen 2016). In essence, al-Maturidi built his theological system on the notion of "hikmah", divine wisdom (Ali 1963, p. 265).

Because of his rationalist epistemology, al-Maturidi on some crucial issues is close to the Mutazilate school (Rahman 2000, p. 62; Ali 1963, p. 273). The assumptions holding that, in the Maturidite theology, reason is subservient to scripture are not true. It is a result of a fundamental misunderstanding of Maturidism (Deen 2016). As such, the above-mentioned two principles show the Maturidite pivotal principle, which accepts that if the real purport of revelation is correctly understood, there can be no conflict between reason and revelation (Ali 1963, p. 264).

By the tenth century, the Hanafi school, which originally was one of the most reason-based Islamic schools and exemplified the "ray" school (reason-based jurisprudence), as a result of strong influence of conservative Asharite teachings and the Ahl al-Hadith school, submitted to the forms of "hadith" (Melchert 1997, p. 31). Although it is claimed by many traditionalist Islamic scholars that Asharism has its own doctrine of rationalism, in fact, it appears that this school espoused a theological position which was critical, if not detrimental, to rationalism (Zhussipbek and Nagayeva 2019, p. 348). Submission of Hanafisim to the forms of "hadith" became the eventual victory of traditionalists. Subsequently a role for reason in mainstream Sunni Islam turned out to be confined within strict limits, mostly in relation to secondary issues of jurisprudence (Hunter 2009, p. 25).

The lack of rationalism in contemporary Hanafite Islamic tradition can also be attributed to the influence of the theological legacy of Hanafite scholars, who introduced traditionalistic Asharite epistemology to Hanafism, such as al-Tahawi, the Hanafi scholar from current-day Egypt. The difference between theological methods of al-Maturidi and al-Tahawi is quite evident. While al-Maturidi was a thorough dialectician who employed rationalistic methods to find out a philosophical basis for his views, al-Tahawi was a true traditionist, who did not favor any rational discussion or speculative thinking on the pillars of faith, accepting them without any questioning. As such al-Tahawi's system may be conceptualized as dogmatic, while that of al-Maturidi as critical. Although both scholars belong to the same Hanafi school, they considerably differ in epistemology and trends of thought (Ali 1963, pp. 245–46).

In the early twentieth century, a prominent reformist, Musa Jarullah Bigiev, who was a Hanafite-Maturidite scholar[2] (Bigiev indicated this fact in his letters, he also performed a pilgrimage and did "itikaf", spiritual retreat, at Abu Hanifa's tomb in Baghdad (Bigi 1975, p. 13)), developed his inclusive interpretation based on the above-mentioned rationalistic Maturidite principles. Rationalistically interpreting Quranic verses (foremost the verses 39:53, 7:156, 20:5), he eloquently advocated religious inclusivism (Bigiif 1911, pp. 47–53). Specifically, the rationalistic principles of compatibility of reason and faith induced him to formulate inclusive Islamic discourse in the book entitled "Rahmat Ilahiyya Burhanlari" (Evidences of God's Mercy).

By employing Maturidite rationalistic principle, Bigiev argued that the belief in eternal punishment of non-Muslims contradicts reason, but in Maturidism reason and faith should not be mutually exclusive. Similarly, to believe that non-Muslims will be sent to Hell forever for their sins committed in this temporary life means the non-acceptance, even denial, of the unlimited mercy of God, which also contradicts human reason (Bigiev 2005, pp. 78, 90–91; Bigiif 1911). On the other hand, from assumption

---

2    Although Bigiev was an un-compromised opponent of "madhab" fanaticism and, on some issues, formulated non-orthodox, nuanced interpretations which may differ from the views of "mainstream" Hanafi-Maturidi scholars.

that belief in eternal punishment for non-Muslims contradicts reason, the argument based on "ethical objectivism" can be inferred. To put it differently, it is ugly and miserable to punish a person forever for the sins committed in this temporary life; however, according to Maturidite theology, God does not order to believe and do what is accepted by reason as bad and evil.

### 3. Analysis of the Theological Principles of Rationalistic Maturidism and Inclusive Religious Interpretations

In our view, in addition to the key rationalistic epistemological principles discussed above, the other Maturidite theological views, such as the Maturidite understanding of the fate of people who did not hear of Prophetic mission, Maturidite interpretations of the foundation of faith and the idea of subjectivity of faith, as well the Maturidite position on predestination, can be seen as the factors to prepare the theological grounds to develop inclusive Islamic interpretations.

### (a) Maturidite views on the fate of people who did not hear of Prophetic mission

There is a difference between Maturidite and Asharite theologies in the issue of the fate of people who did not hear of Prophetic mission. Asharite theology holds that no valid faith can be established among people to whom the Divine mission through Prophets did not reach (in parallel the Asharite theology teaches that only divine revelation can determine what is good and evil, or what is truth and what is not (Hourani 2007, p. 8)). However, based on his emphasis on human reason and intellect, al-Maturidi believed that each human is expected rationally to find her or his Creator (al-Maturidi 1970, pp. 6–11). At the first glance, it may seem that the Maturidite school adopts a stricter stance on issue of the salvation of people who did not hear of Prophetic mission. However, Maturidite school argues that a human being is obliged to have faith in the "Maker". The Maturidite school deliberately uses the concept "Maker" ("sani" in Arabic), but not "Allah". The faith in the "Maker" cannot be the same to the "faith in Allah", as it is conceptualized by Islamic scholars (Matsuyama 2013, p. 4).

As Matsuyama explains, it seems likely that Maturidi scholars deliberately employed in their theology a very general concept "sani" (Maker), but not "khaliq" (Creator), to refer to the One—the belief to whom is to be found rationally by the people who did not hear of Prophetic mission. Moreover, Maturidi scholars intentionally refrained from explaining the reasons why they named God "Maker" (Matsuyama 2013, pp. 4–5). This means that, faith the people who did not hear of Prophetic mission are supposed to have, as such, is not the exclusive faith in "Allah", whom the specific attributes are ascribed to, but inclusive faith in a vaguely conceptualized entity that can be called "Maker" (Matsuyama 2013, p. 5). It is a vital but generally neglected issue in Maturidite theology.

By and large, if to analyze Maturidite theology through a broader perspective, it can be argued that rationalistic Maturidism allows a human being "who is ignorant of Islam" to rationally find faith in one "Maker" and to be saved, although her/his faith is not based on strict Islamic doctrinal formulations (Matsuyama 2013, p. 5). To put it differently, a person without exposure to Prophetic mission in the so-called "Land of Infidelity" can be regarded as a "believer", only if she or he believes in the "Maker" without believing in the specific pillars of the Islamic teachings and without performing any prescribed Islamic obligations. Therefore, in the "Land of Infidelity", it seems not to make sense to distinguish between exclusively-defined Muslims who embrace the concrete teachings of "Islam" (denoting the name of specific religion), and non-Muslims (Matsuyama 2013, p. 7). Hence, the Maturidi position can be interpreted to mean that where Islamic teachings are not known and cannot be known, those who rationalistically came to the belief in one "Maker" ("sani") are to be saved.

Based upon the above-mentioned, we argue that the Maturidite theology may produce the Islamic interpretations which can positively obscure the boundaries between Muslims and non-Muslims, especially between Muslims and adherents of other monotheistic religions not only (Matsuyama 2013, p. 2) in the "Land of Infidelity" (this concept today should be considered "anachronistic"), but also in all places where people have not encountered the mission of "properly-presented Islam" as an exemplary life of a community of believers and the individuals. In this respect, as Matsuyama reminds, it is

undeniable that, despite the backdrop of global revitalization and liberalization of the religious market and diversification of religious discourses, general people in non-Muslim countries have difficulties in having a correct understanding of various aspects of Islamic teachings (Matsuyama 2013, p. 7).

The Maturidite view on the fate of people who did not hear of Prophetic mission can be used to justify the inclusive understanding of Quranic verses (primarily the verses 3:19, 3:85, 5:3) by accepting the lower-case "islam" (denoting the submission to "Maker") instead of upper-case "Islam". The word "islam" (lower case) in these Quranic verses may mean the concept "submission", but not exactly the name of a particular religion ("Islam"). In other words, inclusive reading of the verses: "True religion before Allah is I/islam (submission to His will)" (3:19); "If anyone seeks a religion other than (I/islam) complete devotion to God, it will not be accepted from him ... " (3:85); and "this day I perfected your religion for you, completed My favour to you, and chose I/islam to be your faith" (5:3) allows the word "islam" to be conceptualized as lower-case (not upper-case "Islam"), implying submission or devotion to "Maker" but not exactly the name of the specific religion ("Islam"). The general meaning of the scripture does not impede this kind of inclusive reading (Kamali 2011, p. 714).

Toshihiko Izutsu also argues that the concept "islam" in essence means determined self-surrender and self-submission to the Divine Will (Izutsu 2002, p. 217). The concept "islam" (and the related verb "aslama") basically means that a person puts her/his trust totally in God fully and voluntarily surrenders oneself to the Divine Will, which after all is a kind of unconditional self-surrender (Izutsu 2002, p. 217).

It can be argued that the Asharite theology, despite the fact that it accepts that people who did not hear of Prophetic mission are not responsible to have belief, in fact, is less inclined to religious inclusivism. Moreover, in our days, the proponents of this Asharite principle may argue that in view of the global spread of information about Islam, there is no one left in the world to whom the message of Prophet Muhammad has not yet reached. However, Maturidite theology, by arguing that any human being, as a rational creature, is obliged to rationally find faith in the "Maker", does not confer the obligation to specify the details of this belief and attributes of the "Maker". As such, it only refers to the "Maker" as the "Creator of the World" and never calls "Maker" "Allah" (Matsuyama 2013, p. 5). On the whole, from Maturidite theology it can be inferred that people who do not know Islam may have valid faith which is not characterized by strict Islamic doctrinal formulations, and, accordingly, they will be saved. Furthermore, in line with al-Maturidi's logic (derived from his belief in the necessity of rational belief in the "Maker"), the concept "islam" can be understood primarily as submission (lower case "islam") but not as the exact name of a particular religion. Hence, we argue that the Maturidite logic may be interpreted as meaning that people who believe only in the "Maker" are accepted to be "believers", even if they do not believe in Islamic teachings in a strict sense.

In other words, we argue that, from rationalistic Maturidite theology, it can be inferred that anyone who rationally comes to the belief in the "Maker" (which can be conceptualized as "a mostly correct sense of the nature of the world"[3] is as good as Muslim in God's eyes (as we understand the Maturidite logic, "a mostly correct sense of the nature of the world" is to accept the oneness of the Creator). According to Maturidite theology, "true religion" is the religion of Oneness of God (called "islam") and it never changes, which is distinct from the institutionalized religion associated with the name of Muhammad.

To conclude, first, by recognizing the primary role of human reason in having faith and not confining faith to specific attributes (accepting that the people may reach faith which is similar to valid through their reason, and they may be saved hereafter, although they do not believe in Islamic teachings in a strict sense), it can be argued that Maturidite logic accepts that the religious beliefs that incompatible (lower-case "islam") with narrowly-defined "Islam" (upper-case) may lead to salvation.

---

[3]    We thank an anonymous reviewer who suggested this expression.

Therefore, in this sense, this Maturidite understanding may be seen as laying the foundations to develop an inclusive perspective in Islamic teachings.

**(b) The foundation of faith and subjectivity of faith**

Maturidite theological views on faith can also be used to justify our arguments about the possibilities to develop religious inclusivism based upon rationalistic Maturidite theology.

Al-Maturidi advocated the idea of subjectivity of faith according to which faith is purely an individual assent to God (Basaran 2011, p. 48). The Maturidite position accepts that belief consists only in conviction in the heart (al-Maturidi 1970, pp. 57, 373), whereas Abu Hanifa, whose views al-Maturidi systematized, conceptualized that the foundation of belief consists in conviction in the heart and affirmation by the tongue. However, three other Sunni-Islamic schools and their respective scholars led by Malik, al-Shafii, and Ibn Hanbal accepted that faith also consists of "practice with the limbs" (Haddad 2015, p. 141). As such, al-Maturidi regarded faith as an act of heart and reason, but in its essence, it is affirmation and, in a certain degree, is knowledge (al-Maturidi 1970, pp. 380–81).

Regarding how to come to faith, al-Maturidi advocated an assent that is provided rationally and carried out by the heart ("act of heart"); therefore, faith can be seen as a process rather than a destination (Basaran 2011, p. 54). By emphasizing that belief is an "act of heart", al-Maturidi argued that belief cannot be possible in the presence of force (al-Maturidi 1970, pp. 380–81). Additionally, Al-Maturidi discussed whether belief may be called an act of knowledge ("al-marifa").

The idea of subjectivity of faith was eloquently expressed by Ibrahim Haqqi Erzurumi in the following statement "the fact of various beliefs of various persons is evidence for the greatness of the Creator, because He shows Himself different to all of His servants" (Basaran 2011, p. 54). Moreover, in contradiction to other Islamic schools of doctrine, Maturidis accept that belief does not increase or decrease (al-Maturidi 1970, p. 119), only the degrees of certainty and affirmation may increase or decrease (en-Nesefi 2010, pp. 65–66). Maturidite theology, accepting that actions are not part of belief, maintains that belief is not decreased through sins. Therefore faith can survive sin; consequently, even the worst sinner cannot be treated as an unbeliever (Leaman 2008, p. 86). As Leaman highlights, Maturidite theology, assuming a clear division between faith and human actions, tolerates a serious backsliding and offers rather lenient and relaxed criterion for membership to the religious community (Leaman 2008, pp. 86, 89).

In view of above mentioned theological views on faith, in Maturidite theology, "Islam" (religion) is not indispensable to "iman" (belief), the deeds are not part of the faith. However, not only radical Muslims, but also the conservative traditionalist Sunni Muslims (who strictly follow Asharite theology), adhere to the opposite view and they may accept a believer not performing obligatory deeds, at worst, as an apostate, the view which deeply contradicts rationalistic Maturidite theology.

Furthermore, the Maturidite theological principles assuming that faith in general terms ("bi al-jumlah"), without knowledge about its specific teachings, are principally acceptable seem to be important to develop inclusive Islamic interpretations. As well, according to the Hanafi-Maturidi school, even believers in polytheism are accepted as becoming Muslims simply as a result of their confirmation that "I believe in Allah" (Matsuyama 2013, p. 6).

**(c) Maturidite view on predestination**

We argue that the Maturidite view on predestination seems to be important to infer the foundations of the phenomenon, which can be tentatively called "Islamic individualism", the notion which is vital to develop a pluralistic culture.

Although both Asharite and Maturidite doctrines maintain that human beings are responsible for their own actions, the pivotal notions developed by these schools, ("kasb" (acquisition) by the former school and "ikhtiyar" (choice) by the latter (Lucas 2006, p. 809)), in fact, lead to substantial differences in understanding of free will and predestination, which affect individual and social life.

Al-Ashari taught that human responsibility for her/his own acts depended on acquisition, but he did not accept that a human being is a real actor (Stefon 2010, p. 138). The radical Asharism maintains that God not only creates the actual act but also the individual's will and power (Lucas 2006, p. 809). Asharite theologians, by emphasizing the divine omnipotence at the expense of freedom of will of a human being, developed the theological position which, in substance, is excessively deterministic (Shah 2006, p. 640). Although, to satisfy believers in an intelligible divine justice, al-Ashari developed his theory of justice through the notion of "iktisab" (acquisition), the problem of divine justice was only dismissed but not solved. Since, in his doctrine, it appears that ultimately God predestines what act a man chooses (Hourani 2007, p. 8).

The Maturidite main principle is the idea of cooperation between the Creator and His creation. God creates ("khalq") human actions and human beings do them ("fiil") (Rudolph 2015, p. 305). In other words, al-Maturidi's view on predestination holds that God is the Creator of human acts, although a human being possesses her/his own capacity and free will to commit this act. A human being can; therefore, be seen as the true author of her/his acts; however, in the case of evil acts, they do not occur with the pleasure of the Creator (Shah 2006, p. 640). It can be argued that al-Maturidi developed a doctrine which accepts that a human being is a real actor, though God is the sole Creator of everything. Specifically, some later Maturidi scholars taught that while God creates the act, human beings add specifics, which are uncreated qualifications, to it (Lucas 2006, p. 809). In other words, acting and creating are distinct types of activity pertaining to different aspects of the same human act (Stefon 2010, pp. 138–39).

To conclude, concerning free will and predestination, in comparison with Asharism, the Maturidite theology, by developing the doctrine of "kasb", allots human beings the power of real actor and responsibility for her or his choices. In essence, in Maturidite doctrine, the initial choice is of a human, whereas in al-Ashari teaching it is of God (Halverson 2010, p. 27). As such, Asharite position on free will makes inconceivable the idea that a human being is individually responsible for her or his acts (Thiele 2016, p. 228). Bigiev heavily criticized the excessively deterministic views on predestination inspired by traditionalistic scholarship and argued that the not properly understood notion of fate and interpretations of predestination led the Muslims to poverty, underdevelopment, and depression (Bigi 1975, p. vii).

To conclude, the Maturidite view on predestination may provide a ground to develop "Islamic individualism". Individualism is a key factor in developing pluralism and pluralistic culture, which lays the foundations of inclusive religious interpretations. In general, respect for individual freedom is one of the core liberal values and fundamentals of democratic culture.

**(d) Acceptance of irja**

On the other hand, that al-Maturidi accepted and defended the notion of "irja" (literally means "pushing back of judgment", the acceptance that all faiths are to be judged by God on the Day of Judgement)[4] can also be used to discuss the potential of Maturidite theology to produce inclusive Islamic understandings. As Rudolph emphasizes, although al-Maturidi did not accept being called "Murjiite", he rehabilitated the "Murjiites" as long as the term was understood to refer to the adherents of the "true understanding" of "irja" (al-Maturidi explained that "God alone knows people's hearts and knows who believes and disbelieves. People and even angels are not capable of judging. Whoever claims to do so despite this, is committing disbelief". The angels first "pushed back" their judgment ("irja") and were a role model for everyone through this (see in Rudolph 2015, p. 51)), thereby he finds a way, despite his probable unease, to acknowledge this characteristic in his own tradition (Rudolph 2015, p. 155).

---

4    We are grateful for an anonymous reviewer who draws attention to al-Maturidi's discussion of "irja" in our conceptualization of the potential of inclusiveness of Maturidite principles.

We argue that al-Maturidi, like Abu Hanifa, accepted and defended the notion of "irja" as a concept derived from Quran to be used as a rationalistic method of understanding religion (Erdem 2012, pp. 145–46). For example, al-Maturidi emphasized that "Even if a hadith says that a sinner is no longer a believer, this is not correct. The hadith must be wrong, since it contradicts the Quran, and its transmitter is blameworthy" (see in Rudolph 2015, p. 51).

## 4. Conclusions

In this paper we aim to search for the theological grounds to produce inclusive Islamic interpretations in largely-lost-for-centuries Islamic rationalistic tradition, by which we depict a religious school named Maturidism. While the views of al-Maturidi per se cannot be categorized as inclusive (the limits of al-Maturidi's inclusiveness should be acknowledged), Maturidite epistemology and some key Maturidite views can be used to lay the grounds to develop inclusive Islamic discourses. As such, it is necessary to differentiate between Maturidite "software" and "Maturidite" hardware, where the former means dynamic rationalistic Maturidite theology, the latter denotes traditional Maturidite creed and static, even "ossified", Maturidite "kalam" (Zhussipbek and Nagayeva 2019, p. 354).

By and large, there ought to be an incentive to go from exclusivist al-Maturidi views to inclusivism-inclined Maturidite rationalistic principles. Overall, there is a nexus between the theologically rationalistic approach to Islamic sources and the development of inclusive religious interpretations. Specifically, the rationalistic principle of compatibility of reason and faith can be used to develop inclusive Islamic discourse. The reformist Maturidite scholar, Musa Bigiev, by arguing that the belief in eternal punishment of non-Muslims contradicts reason, unambiguously employed this principle to develop his inclusive Islamic theory in the early twentieth century.

Furthermore, the Maturidite view on the fate of people who did not hear of Prophetic mission may be understood as accepting that the people may reach faith, which is similar to valid through their reason, and they may be saved hereafter, although they do not believe in Islamic teachings in a strict sense and do not necessarily live in the "Lands of Infidelity" per se, but in all places where they have not been exposed to the mission of "properly-presented Islam" as exemplary life of a community of believers and the individuals. In other words, the religious beliefs of many people (if not of all non-Muslims) in today's world that are incompatible with the narrowly defined "Islam" (upper-case) may lead to salvation; therefore, this Maturidite understanding may be seen as laying the foundations to develop inclusive discourse.

On the other hand, the Maturidite logic can also be used to substantiate the arguments to interpret and read the Quranic verses inclusively, to accept lower-case "islam" as denoting the submission to "Maker", instead of upper-case "Islam". In other words, inclusive reading of the Quranic verses (3:19, 3:85, 5:3), which may be inferred from Maturidite view on people not exposed to Prophetic mission, closes the door of religious exclusivism.

On the whole, by invoking the principles of rationalistic Maturidite theology and Maturidite theological views on the fate of people who did not hear of Prophetic mission, foundation of faith, and predestination, it is possible to develop inclusive Islamic discourses by using internal Islamic dynamics.

To summarize, we argue that from rationalistic Maturidite theology it can be inferred that anyone who rationally comes to belief in the "Maker" is as good as Muslim in God's eyes. In other words, the logic assuming the rationalistic position that God is knowable by natural means, as it was conceptualized by al-Maturidi, can be used to justify inclusive Islamic interpretations in our days. In other words, in view of the fact that the rationalistic position assuming that God is knowable by natural means is the cornerstone of al-Maturidi's entire intellectual edifice, and the acceptance of a human's rational capacity occupies a central position in his definition of the human being (Rudolph 2015, p. 300), we can argue that, if to follow this logic, ascribing to human reason such capacity, the development of inclusive interpretations of Islam in our days is possible.

**Author Contributions:** Both authors contributed to the general conceptualization and methodology. 1st author (G.Z.) analyzed al-Maturidi's epistemology, his theological views in Kitab al-Tawhid, and Maturidite views on predestination. 2nd author (B.S.) analyzed the foundations of belief in Maturidite theology, the notion of irja and reformist Maturidite scholars like Musa Bigiev.

**Funding:** This research was funded by the research grant provided by the Committee of Science, Ministry of Education and Science, Republic of Kazakhstan, grant number is AP05133414. The APC was funded by the same grant.

**Conflicts of Interest:** The authors declare no conflict of interest.

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
