# Peer review of "Search for the Theological Grounds to Develop Inclusive Islamic Interpretations: Some Insights from Rationalistic Islamic Maturidite Theology"

_religions, doi:10.3390/rel10110609_

Round 1

Reviewer 1 Report

Maturidi’s Kitab al-Tawhid is available in two editions (ed. Fath-Allah Khulayf, Cairo, 1970 & eds. Tubaloglu, M. Arushi, Beirut and Istanbul, 2002), both available for download online

https://archive.org/details/Kitab_tawhid_Materidi

There is no reason why this does not become the main source of the claim that Maturidi has developed a theory of inclusiveness in his theology.

The limits of Maturidi’s inclusiveness also need to be stated clearly. A broad acknowledgement of these limits can be found in lines 62-4 but not explained.  [Although, Al-Maturidi’s views per se about other religions can hardly be categorized as inclusive, nonetheless, we argue that rationalistic Maturidite theology, specifically its two principles, which are briefly touched in this article, can be used to produce inclusive Islamic understandings.]  Maturidi argues against the determinists (ashab al-taba’i‘---modern time equivalents would be the followers of Stanford’s Robert Sapolsky), dualists/Zoroastrians, Buddhists (sumaniyya), sophists, and while he is willing to employ the agreement of Muslims with other monotheists (such as Jews and Christians who make small mistakes in understanding God), he still argues against Jews and Christians.  He is an inclusivist, I think, in two important meanings (though the authors do not have to agree with me on this):

He allows all faiths to be judged by God in the Ultimate Day (in his discussion of irja’, or delay of judgment) He genuinely as this article argues (following Matsuyama and others) thinks that those who settled on a mostly correct sense of the nature of the world and its benevolent maker are as good as Muslims in God’s eyes

Some sentences need to be edited for clarity and grace, but leave this to the very end, after the text of the article itself is finalized.

I am not calling for extensive changes.  These changes can be made in a matter of one week, if the authors use Maturidi’s Tawhid pragmatically and focus on the relevant points.

Author Response

We accept all comments of our reviewer and made relevant revisions in our manuscript. We renamed 2nd sub-section and included the refutations of some Islamic sects and other religions made by al-Maturidi in his book Kitab al-Tawhid. Also we added some new information about irja. Finally we sharpened our views (in conclusion) about the potential of rationalistic Islam to produce inclusive interpretations. 

Reviewer 2 Report

I would suggest to emphasize a bit more that here an attempt of a new interpretation is made which is based on Maturidi's principles but in contradiction with what he himself really wrote.

Author Response

We accept the comments of our reviewer. We made more explicit the exclusivist religious views of al-Maturidi by referring to his book Kitab al-Tawhid. Also we included some new information and interpretations to show the potential of rationalistic Islam (in distinction to al-Maturidi's views) to produce inclusive Islamic interpretations. 

Round 2

Reviewer 1 Report

Reliance on primary sources remain meager.